# Antimicrobial Defined Daily Dose in Neonatal Population: Validation in the Clinical Practice

**DOI:** 10.3390/antibiotics12030602

**Published:** 2023-03-17

**Authors:** Cristina Villanueva-Bueno, Elena Montecatine-Alonso, Francisco Jiménez-Parrilla, María González-López, Silvia Manrique-Rodríguez, Francisco Moreno-Ramos, Carme Cañete-Ramírez, Elisenda Dolz, Ana García-Robles, José Manuel Caro-Teller, María Teresa Moral-Pumarega, Elena Bergon-Sendin, María Teresa Gómez-Trevecedo Calvo, Carmen Gallego-Fernández, Concepción Álvarez del Vayo-Benito, Marta Mejías-Trueba, María Victoria Gil-Navarro

**Affiliations:** 1Department of Pharmacy, Hospital Gregorio Marañón, Instituto de Investigación Sanitaria Gregorio Marañon, 28007 Madrid, Spain; 2Department of Pharmacy, Hospital Universitario Virgen del Rocío, 41013 Sevilla, Spainmartamejiastrueba@hotmail.com (M.M.-T.);; 3Department of Neonatology, Hospital Universitario Virgen del Rocío, 41013 Seville, Spain; 4Department of Neonatology, Hospital Regional Universitario de Málaga, 29010 Málaga, Spain; 5Department of Pharmacy, Hospital Universitario La Paz, 28046 Madrid, Spain; 6Department of Pharmacy, Hospital Universitario Vall d’Hebron, 08035 Barcelona, Spain; 7Department of Pharmacy, Complejo Hospitalario Universitario Insular-Materno Infantil, 35016 Las Palmas de Gran Canarias, Spain; 8Division of Pharmacy, Hospital Universitario y Politécnico La Fe, 46026 Valencia, Spain; 9Department of Pharmacy, Hospital Universitario 12 de Octubre, Research Institute 12 de Octubre i+12, 28041 Madrid, Spain; 10Department of Neonatology, Hospital Universitario 12 de Octubre, 28041 Madrid, Spain; 11Department of Pharmacy, Hospital Universitario de Jerez, 11407 Jerez de la Frontera, Spain; 12Department of Pharmacy, Hospital Regional Universitario de Málaga, 29010 Málaga, Spain; 13Department of Infectious Diseases, Microbiology and Parasitology, Institute of Biomedicine of Seville, University Hospital Virgen del Rocío, Spanish National Research Council, University of Seville, 41013 Seville, Spain; 14Institute of Biomedicine of Seville, Virgen del Rocío University Hospital, CSIC, University of Seville, 41013 Seville, Spain; 15Centro de Investigación Biomédica en Red de Enfermedades Infecciosas, 28029 Madrid, Spain

**Keywords:** antimicrobial consumption, neonatal antimicrobial prescription, neonatal infections, antimicrobial management, daily-defined dosage

## Abstract

Background: Currently, there is no validated method for estimating antimicrobial consumption in the neonatal population, as it exists for adults using Defined Daily Doses (DDD). In neonatology, although there are different methods, each one with advantages and disadvantages, there is no unified criterion for use. The aim of this study is to validate the neonatal DDD designed as a new standardised form of antimicrobial consumption over this population. Methods: The validation of the neonatal DDD, Phase II of the research project, was carried out through a descriptive observational study. Periodic cut-offs were performed to collect antimicrobial prescriptions of neonates admitted to the neonatology and intensive care units of nine Spanish hospitals. The data collected included demographic variables (gestational age, postnatal age, weight and sex), antimicrobial dose, frequency and route of administration. The selection of the optimal DDD value takes into account power value, magnitude obtained from the differences in the DDD, statistical significance obtained by the Wilcoxon test and degree of agreement in the stipulated doses. Results: Set of 904 prescriptions were collected and finally 860 were analysed based on the established criteria. The antimicrobials were mostly prescribed in the intensive care unit (63.1%). 32 different antimicrobials were collected, and intravenous administration was the most commonly used route. Neonatal DDD were defined for 11 different antimicrobials. A potency > 80% was obtained in 7 antibiotics. The 57.1% of the selected DDD correspond to phase I and 21.4% from phase II. Conclusion: DDD validation has been achieved for the majority of intravenously administered antimicrobials used in clinical practice in the neonatal population. This will make it possible to have an indicator that will be used globally to estimate the consumption of antimicrobials in this population, thus confirming its usefulness and applicability.

## 1. Introduction

Antimicrobials are one of the most prescribed groups of drugs in neonatology, especially in intensive care units (NICU), as shown by numerous studies on their use in this population [1]. Factors such as the high diagnostic-etiological uncertainty or the prematurity of neonates lead to abuse and inappropriate use of antimicrobials [2,3]. 

The results of the latest Global Point Prevalence Survey on Antimicrobial Use and Resistance at Hospital Level (Global-PPS) [4], show the causes associated with inadequate antimicrobial prescribing: high use of certain classes of antibiotics (62%), prolonged surgical prophylaxis (60.9%), lack of review of discontinuation date (51%), empirically used antimicrobials (43%) and frequent use of multiple antibiotics per indication/patient (22%). As well as similar data were found in a recent multicentre, cross-sectional study on the prevalence and appropriateness of antimicrobial use in Spanish hospitals [5] shows that antimicrobial prescribing was adequate in 34% of cases, improvable in 45% and inadequate in 19%. The main causes of inadequacy were choice of agent, duration of treatment and lack of monitoring of efficacy and safety. 

Considering that the use of antimicrobials is not without consequences, such as changes in the microbiota in both microbial community composition and antimicrobial resistance gene profile [6,7,8] or increased risk of morbidity [9,10,11,12,13], knowledge of antimicrobial prescribing and consumption data is key to optimizing their use. Therefore, it is essential to be able to analyze trends in the prescription and use of antimicrobials, thus addressing a growing health problem: the generation of antimicrobial resistance [14]. 

In this regard, for adults, the defined daily dose (DDD), established by the World Health Organization (WHO) as the average standard daily dose of a medicine used in a 70 kg adult for the most common indication, is one of the universally used metrics to assess antimicrobial consumption. However, the validity of the WHO definition of DDD is questionable in neonatology, where drug dosing is based on body weight [15]. 

A systematic review found up to 26 different measures in 79 studies (DDD or similar metrics in 38 studies). Of all of them, only 47 focused on the pediatric and neonatal population [16]. Among the different measures available, the following stand out: Point Prevalence Surveys (PPS) that can evaluate the consumption of antimicrobials in short periods [17]. However, data are susceptible to the complexity of case mix, seasonality and sample variability. Days Of Therapy (DOT) [18], the number of days, a patient receives a given antibiotic, it minimises the impact of dose variability (useful in paediatrics and in patients with renal failure) but it requires a complex and variable calculation. Prescribed Daily Dose (PDD), the usually prescribed dose of a given antibiotic, it is a closer approximation of the doses used. Nevertheless, it is a non-standardised method, and it is difficult to establish comparisons between centres.

In short, there is currently no standardized method for the neonatal population and the different options available have different advantages and disadvantages. For this reason, it is necessary to have a global metric that can be easily calculated and that allows comparisons to be made between the different centres. In view of the need, a useful method for antimicrobial DDD measurement in neonates was designed [19]. For this purpose, a multicentre observational study was carried out to obtain the theoretical DDD. Therefore, the main objective of this study is to validate the theoretical DDD obtained from the method developed for use in the neonatal population and to establish the most appropriate DDD values to be used in clinical practice in this population.

## 2. Materials and Methods

### 2.1. Study Design

This is an observational, retrospective, multicentre study consisting of two phases.

The Phase I was aimed at the theoretical calculation of neonatal DDD. For this purpose, the mean weight of 4820 neonates admitted into six Spanish hospitals was considered, and the doses for each antimicrobial in its most common indication were agreed by consensus using the Delphi method. DDD (g) for each antimicrobial was calculated, by multiplying the total average weight of the cohort (kg) and the recommended dose for the most common indication of each antimicrobial (mg/kg) previously agreed. Neonatal DDD were designed for 47 antimicrobials (31 administered intravenously and 16 orally). The results of this first phase have been published elsewhere [19].

The Phase II, detailed in this manuscript, constitutes the validation process of the DDD designed in the Phase I. Antimicrobial prescriptions were collected from neonatal wards of 9 Spanish hospitals over a 2-year period (March 2019 and May 2021).

### 2.2. Data Collection

Prescriptions were collected with weekly or fortnightly prevalence cut-off. A researcher from each hospital collected the data through electronic prescription systems, medical records or paper format, depending on the systems and technical support available at each centre. All variables studied were stored in an electronic data collection notebook: demographic variables (gestational and postnatal age expressed in weeks and days, sex and weight) and antimicrobials used (active substance, dose, frequency and route of administration). 

From the data collected, the total dose of antibiotic received per patient (mg/day) was calculated and subsequently, the median of the resulting DDD per antibiotic (g/day) was obtained. These DDD obtained from actual prescriptions (Phase II) were compared with the theoretical DDD (Phase I). 

### 2.3. Data Analysis

Non-parametric methods were used. Median and interquartile ranges (IQR) were estimated in quantitative variables and frequency and percentage in qualitative variables. 

To describe the DDD of each antibiotic obtained in Phase II, the median and its population confidence intervals at 95% (CI95%) were used, and the difference observed between phases was analysed using the Wilcoxon nonparametric test. The power value for each Wilcoxon test was performed and the p-value was also calculated. 

For all inferential statistical calculations, an alpha error value of 5% and a beta error of 20% were used. RStudio 2022.02.3+492 “Prairie Trillium” Release for Windows was used to perform the statistical analyses.

### 2.4. DDD Selection Criteria

The selection of the optimal DDD value takes into account: the power value, the magnitude obtained from the differences in DDD medians between phases, the statistical significance obtained by the Wilcoxon test and the degree of agreement with the dose used for the DDD calculation in the first phase.

Antimicrobials with a potency greater than 80% were evaluated. Of these, phase I or II DDDs were selected based on the results of the Wilcoxon test and the magnitude of the differences in the DDDs between both phases, fundamentally. A high degree of consensus on the dose used (≥75%) was also positively assessed, although it was considered a conditional criterion. For the remaining antimicrobials whose power was less than 80%, statistically we cannot state if there are differences between phase I and phase II DDD, but for those where there was a consensus of ≥75% we could use phase I DDD with caution. The selection criteria are shown in Table 1.

### 2.5. Ethical Approval

This study was a completely anonymous audit of current antimicrobial prescribing practice. No unique identifiers obtained. The study was approved by the Spanish Agency for Medicines and Sanitary Products (ID number: GAT-TEI-2015-01) and by the Ethics Committee of the Virgen Macarena and Virgen del Rocío University Hospitals (identification number: 0620-N-15) and also has the authorisations of the ethics committees of each of the participating centres.

## 3. Results

A total of 904 prescriptions were collected. Of these, 860 prescriptions were analysed after eliminating those that did not meet all the criteria or did not show the requested variables. A first descriptive analysis of all the data shows that the prescriptions analysed were similarly distributed among the three groups of neonates established according to gestational age, immature (35.6%), preterm (29.35) and term (35.1%). The NICU was the unit where the highest number of prescriptions was collected. 60% of the neonates were more than 7 days old, with a mean weight of 2.260 kg. Regarding the sex variable, a greater number of prescriptions were collected from male neonates (57.4%) (Table 2). The demographic characteristics of the patients whose prescriptions were analysed show that 63.14% of the patients requiring antimicrobial therapy were admitted to the NICU, and it is also in these units that the neonates with the lowest gestational age (46% were younger than 29 weeks) and lowest mean weight (1.600 kg) were admitted (Table 3).

Thirty-two different antimicrobials were prescribed, and the intravenous route of administration was the most commonly used (96.2%). According to the antimicrobial prescription profile, the distribution of antimicrobials by hospital units shows: 63.1% (*n* = 543) of prescriptions belong to NICU, 28.9% (*n* = 249) to neonatology and 7.9% (*n* = 68) to neonatal intermediate care. The five most prescribed antimicrobials were ampicillin (23.4%), gentamicin (21.6%), vancomycin (13.6%), cefotaxime (8.4%) and amikacin (9.5%).

Table 4 shows the DDD value and the degree of consensus on the dose used to calculate the Phase I DDD, the median of the DDD resulting from Phase II with its CI95% for each antimicrobial and the differences between the DDD Phase I and II. The value of the DDD finally selected is also showed. A total of 7 out of 14 antimicrobials had a power >80%. Of the remaining antimicrobials with ≤80% potency, the phase I DDD was used in four of them due to the high degree of consensus. In three antimicrobials (cefazolin, teicoplanin, and piperacillin/tazobactam) it was not possible to define the DDD due to the lack of power and consensus in the panel of experts.

## 4. Discussion

In this study we analysed the validity of the Phase I DDD designed to estimate antimicrobial consumption in real clinical practice. Furthermore, thanks to the collection of antimicrobial prescriptions, we corroborated that both the demographic characteristics of the neonates included in the study and the use of antimicrobials analysed resemble those described in the literature on the neonatal population.

In terms of antimicrobials prescribed, the study by Al-Turkait et al. [1] lists the 100 most prescribed antimicrobials in NICUs worldwide, with ampicillin, gentamicin, vancomycin, cefotaxime and tobramycin being the top five. These antibiotics are in line with antimicrobial consumption trends shown in numerous European publications [20,21] and are some of the most prescribed antimicrobials in our study. In fact, in our research we were able to validate the value of neonatal DDD for four of the five top antimicrobials. Even the combination of aminoglycosides with beta-lactams, the most commonly used in neonates with sepsis or suspected sepsis according to published data [22], coincides with the data for the two antibiotics (gentamicin and ampicillin) with the highest number of prescriptions in hospitalised neonates in our study centres [23].

Our study includes 904 antimicrobial prescriptions in neonatal units and NICUs in 9 hospitals throughout Spain. Internationally, reviews have been published on the different methodologies used to determine antimicrobial consumption in the neonatal and paediatric population. In this regard, Rosli et al. [24] identified a total of 20 drug utilisation studies, 8 of which focused on antimicrobials. The ATC/DDD drug consumption reporting system appears in studies evaluating antimicrobials. It is the most commonly used methodology when the aim is to compare antimicrobial consumption between centres. However, DDD in neonatology is not defined or validated and no ideal method has been established for the neonatal population. Therefore, studies have been conducted with the aim of developing a new standardised way of comparing antimicrobial prescribing rates. Porta et al. propose a new pragmatic three-step algorithm carried out in four children’s hospitals in three European countries (UK, Greece and Italy). The first step includes a simple comparison of the proportion of children hospitalised with antibiotics by weight bands and the number of antimicrobials accounting for 90% of total DDD drug use, the second step is a comparison of the dosing used and the third step is to compare overall drug exposure using DDD/100 bed days for standardised weight bands between centres. Although they collected 1217 prescriptions, only 21% were in the neonatal population (263/1217). Liem et al. [25] defined neonatal DDD for ten antimicrobials but failed to validate it.

A particularly relevant consideration is highlighted in the research work of Channon-Wells et al. [26], where they observed that the data source was a more important source of variation than the reported metric (DDD versus DOT). It was also observed that, in most cases, DDD and DOT were highly correlated. In our study, we focused on DDD as a method of consumption because of the many advantages it offers, especially the rapid collection of data and the existence of databases needed to calculate consumption in all healthcare centres, which are clearly the limiting factors according to studies measuring antimicrobial consumption. In Europe, there are few standardised electronic prescribing databases in hospitals that provide accurate information for complex calculations of antimicrobial use. However, all hospitals are accustomed to calculating DDD for adults, so the proposed methodology for the neonatal population would be easy to adopt and calculate by centres after minor adaptation.

The calculation of neonatal DDD in our study is based on the DDD model defined by the WHO for adults and is designed according to two key variables, weight and dosage.

Regarding dosage, the doses were agreed upon by a multidisciplinary group of experts including hospital pharmacists, neonatologists, paediatricians and infectious disease specialists, and follow both national recommendations and international guidelines, which would allow their adoption in other countries.

In terms of weight, Phase I (DDD Design) established a median of 2.81 kg meanwhile in Phase II (DDD Validation) the value decreased up to 2.26 kg. At the national level, a Spanish cross-sectional growth study published in 2008 shows the weight values of Caucasian live newborns of 26–42 weeks gestational age born between 1999 and 2002 in two Spanish hospitals. Thus, for the mean gestational age of Phase I of our study, 36.72 weeks, Carrascosa et al. [27] reported a mean weight between 2.639 kg (36 weeks gestational age) and 2.904 kg (37 weeks gestational age). Other European studies have found similar weight ranges for this population: from 2.0 kg (Liem et al.) to 2.7 kg (Distrasakou et al.) [28,29,30,31]. Although the median weights were slightly different between the two phases of our study, both are within the weight ranges of national and European studies. The asymmetric distribution of the patient profile (immature patients represent 34.7% of the included neonates) may explain the deviation from the estimated DDD in phase I for certain antimicrobials. The actual data collected clearly show that neonates admitted to intensive care units are those of lower gestational age and therefore lower birth weight. Therefore, it is not surprising that most of the antimicrobial prescriptions were made in the NICU due to the more severe and unstable situation of these patients (multiple comorbidities due to organ immaturity), which is in line with standard clinical practice.

From a statistical point of view, the analysis of neonatal DDD shows, on the one hand, that the antibiotics that achieved a greater degree of consensus in the established dose to calculate the theoretical DDD are those that later present greater similarity in the DDD values between both phases. On the other hand, of all the antimicrobials initially selected, we have established a validated DDD for those with the highest number of prescriptions and most commonly used in routine clinical practice, including the antimicrobials whose DDD values were calculated in the study by Liem et al., the only neonatal DDD published to date.

From a clinical point of view, the DDD of vancomycin shows discrepancies. This drug is very susceptible to monitoring and prescription at doses adjusted to pharmacokinetic levels [32,33], which explains the differences as doses can vary greatly from the standard and even vary from patient to patient.

The DDD of cefotaxime and meropenem seem to underestimate their consumption. The use of these drugs occurs in the NICU, where severe infections, such as those caused by *Pseudomonas spp*. or affecting the central nervous system, lead to the use of higher doses than usual. In the case of the clearly discordant antimicrobials (cloxacillin, amoxicillin, linezolid and amoxicillin-clavulanic acid), none have a sufficient number of prescriptions to validate DDD. However, the high degree of consensus achieved among the panelists was considered sufficient criteria to establish the Phase I DDD, which must be further validated in future studies. Moreover, in the case of fluconazole, its use as prophylaxis and not as routine treatment in a high number of cases, prevents validation of the DDD value.

Main limitations of our study include the lack of data collection for some variables (gestational age, postnatal age and income unit). However, this does not preclude the analysis of DDD. Even with the data available for these variables, it allowed us to see trends in the characteristics of patients requiring the use of antimicrobials, mainly preterm infants. Moreover, there are antimicrobials with a low utilisation in routine clinical practice in neonatology. In fact, 68.0% (32/47) of the Phase I antimicrobials could not be analysed due to lack of prescriptions in clinical practice or insufficient number of prescriptions to perform the analysis. Furthermore, only intravenous antimicrobials were analysed because oral antimicrobials (3.8% of the prescriptions collected) were practically not prescribed in the study population, as is usually the case in real clinical practice.

It should be noted that the use of neonatal DDD does not assess the indication; it is only a measure to estimate antimicrobial consumption taking into account the dose for the most common indication. Therefore, it does not accurately reflect doses in situations where antimicrobial monitoring is required and therefore doses adjust according to blood drug levels, clinical situations where patients require higher doses than usual: severe CNS infection, sepsis or otitis. All these situations can also happen in the adult population and, despite this, the use of DDD has been standardised as a method of calculating antimicrobial consumption and is used internationally.

The study has 4820 neonates included in Phase I, and around 900 antimicrobial prescriptions collected from real clinical practice for the validation process. Hospitals with reference neonatal units have participated, including highly specialised intensive care units throughout Spain. In addition, the study involved a multidisciplinary group of experts to reach a consensus on the doses used to calculate the DDD and subsequent clinical analysis during the validation process. The doses agreed by the expert group follow national recommendations as well as international guidelines to be easily adapted at international level.

To the best of our knowledge, this is the first study aimed at designing and validating DDD for the neonatal population. This is highly relevant, since it is necessary to have an indicator adapted to such a complex population. In addition, taking into account the applied methodology, the values obtained can be extrapolated and allow their global implementation. Although there is still uncertainty about the most appropriate metrics in neonates, it must be taken into account that some centres do not have the resources to measure many of them; while obtaining DDDs is plausible in most settings, additional options for newborns are a real need.

## Figures and Tables

**Table 1 antibiotics-12-00602-t001:** DDD selection criteria.

Power	DDD Selection	
>80%	Phase I	Phase II
	There are no significant differences (*p* > 0.01) + Clinical difference magnitude (≤10%) +/− Degree of agreement (≥75%)	Statistically significant differences (*p* < 0.01) + Clinical difference magnitude (>10%)
	Statistically significant differences (*p* < 0.01) + Clinical difference magnitude (≤10%) +/− Degree of agreement (≥75%)	
	There are no significant differences (*p* > 0.01) + Clinical difference magnitude (>10%) +/− Degree of agreement (≥75%)	
≤80%	Degree of agreement (≥75%)	NA

**Table 2 antibiotics-12-00602-t002:** Demographic characteristics of the neonates and the treatments administered.

	Overall (N = 904)
**Income Unit**	
N-Miss	44
Neonatal Intermediate care	68 (7.9%)
Neonatology	249 (29.0%)
Neonatal intensive care Unit (NICU)	543 (63.1%)
**Gestational age (weeks)**	
N-Miss	340
<29	201 (35.6%)
30–36	165 (29.3%)
37–44	198 (35.1%)
Gender	
Female	385 (42.6%)
Male	519 (57.4%)
**Postnatal age (days)**	
N-Miss	232
<7	269 (40.0%)
>7	403 (60.0%)
**Weight (grams)**	
Median	2.260
Q1, Q3	1.098, 3.270
**Antimicrobials administered**	
Liposomal amphotericin B	9 (1.0%)
Amikacin	82 (9.1%)
Amoxicillin	11 (1.2%)
Amoxicillin-clavulanic	9 (1.0%)
Ampicillin	201 (22.4%)
Amphotericin	1 (0.1%)
Azithromycin	1 (0.1%)
Cefazolin	12 (1.3%)
Cefepime	3 (0.3%)
Cefotaxime	72 (8.0%)
Ceftazidime	3 (0.3%)
Cefuroxime	3 (0.3%)
Ciprofloxacin	2 (0.2%)
Clindamycin	3 (0.3%)
Cloxacillin	9 (1.0%)
Daptomycin	3 (0.3%)
Erythromycin	1 (0.1%)
Fluconazole	40 (4.4%)
Fosfomycin	1 (0.1%)
Gentamicin	191 (21.3%)
Linezolid	11 (1.2%)
Meropenem	55 (6.1%)
Metronidazole	1 (0.1%)
Micafungin	1 (0.1%)
Benzylpenicillin sodium	1 (0.1%)
Piperacillin/tazobactam	25 (2.8%)
Rifampicin	1 (0.1%)
Teicoplanin	7 (0.8%)
Trimethoprim/sulfamethoxazole	1 (0.1%)
Vancomycin	138 (15.3%)
**Route of administration**	
Oral	34 (3.8%)
Intravenous	870 (96.2%)

N-Miss: Number of prescriptions that cannot be classified according to the established criteria (unit of entry, gestational age and postnatal age) due to lack of data.

**Table 3 antibiotics-12-00602-t003:** Distribution of neonates in hospital units according to gestational age and weight.

	Neonatology (N = 249)	Neonatal Intermediate Care (N = 68)	NICU (N = 543)	Total (N = 860)	*p* Value
**Gestational age (weeks)**					<0.001
N-Miss	144	30	156	330	
<29	5 (4.8%)	5 (13.2%)	179 (46.3%)	189 (35.7%)	
30–36	43 (41.0%)	12 (31.6%)	102 (26.4%)	157 (29.6%)	
37–44	57 (54.3%)	21 (55.3%)	106 (27.4%)	184 (34.7%)	
**Weight (grams)**					<0.001
Median	2.940	2.048	1.600	2.280	
Q1, Q3	2.020, 3.400	1.000, 3.100	0.985, 3.000	1.100, 3.275	

NICU: Neonatal Intensive Care Unit. N-Miss: Number of patients whose age and weight data were not available.

**Table 4 antibiotics-12-00602-t004:** Neonatal defined daily dose of antimicrobials according to the results of data analysis.

Antimicrobials		Phase I DDD		Phase II DDD		Difference with Phase I DDD				DDD Differences between Phases		Power Value>80%	Difference Value (≤10%)	Wilcoxon Test (>0.01)	Degree of Agreement (≥75%)	Selected DDD	Final DDD (g/day)
	N	Value (g/day)	Degree of Agreement	Median (g/day)	CI95%	Median	CI95%	Power (1-B) %	Wilcoxon Test	Median	CI95%						
Amikacin	82	0.04	100	0.032	0.022; 0.037	−0.001	−0.007; 0.008	99.9	<0.001	−3	−17.5; 20	Yes	Yes	No	Yes	Phase I	0.04
Fluconazole	39	0.02	100	0.002	0.002; 0.004	0.007	0.006; 0.008	99.9	<0.001	35	30; 40	Yes	No	No	Yes	Phase II	0.002
Cefotaxime	72	0.27	80	0.352	0.255; 0.435	−0.122	−0.205; −0.025	94.9	0.003	−45	−75.9; −9.3	Yes	No	No	Yes	Phase II	0.35
Vancomycin	138	0.08	80	0.05	0.04; 0.06	0.02	0.010; 0.030	94.4	<0.001	25	12.5; 37.5	Yes	No	No	Yes	Phase II	0.05
Meropenem	55	0.11	66.7	0.12	0.09; 0.180	−0.03	−0.09; 0.00	91.0	0.026	−27	−81.8; 0	Yes	No	Yes	No	Phase I	0.11
Gentamycin	191	0.01	90	0.009	0.007; 0.010	0.001	0.00; 0.003	85.2	<0.001	10	0; 30	Yes	Yes	No	Yes	Phase I	0.01
Ampicillin	201	0.27	80	0.28	0.25; 0.30	−0.05	−0.07; −0.02	83.0	0.304	−19	−25.9; −7.4	Yes	No	Yes	Yes	Phase I	0.27
Cloxacillin	9	0.13	80	0.28	0.09; 0.400	−0.17	−0.29; 0.02	59.1	0.075	−131	−223.1; 15.4	No	No	NA	Yes	Phase I	0.13
Amoxicillin	10	0.08	75	0.047	0.037; 0.090	0.023	−0.020; 0.033	57.7	0.126	29	−25.0, 41.3	No	No	NA	Yes	Phase I	0.08
Cefazolin	12	0.13	60	0.139	0.0; 0.264	−0.029	−0.15; −0.110	11.3	0.609	−22	−115.4; −84.6	No	No	NA	No	-	-
Linezolid	11	0.08	89	0.06	0.018; 0.105	0.01	−0.035; 0.052	11.2	0.229	13	−43.8; 65	No	No	NA	Yes	Phase I	0.08
Amoxicillin-Clav	9	0.27	80	0.3	0.10; 0.36	−0.07	−0.13; 0.13	11.1	0.905	−26	−48.1; 48.1	No	No	NA	Yes	Phase I	0.27
Teicoplanin	7	0.02	55.6	0.02	0.007; 0.035	−0.01	−0.025; 0.003	7.5	0.999	−50	−125.0; 15	No	No	NA	No	-	-
Piperacillin/ Tazobactam	25	0.54	66.7	0.525	0.21; 0.75	−0.075	−0.30; 0.24	5	0.886	−14	−55.6; 44.4	No	No	NA	No	-	-

DDD: Defined Daily Doses, IQR: interquartile range, CI: Confidence Interval.

## Data Availability

Not applicable.

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
