# Peer review of "Antimicrobial Defined Daily Dose in Neonatal Population: Validation in the Clinical Practice"

_antibiotics, 2023, doi:10.3390/antibiotics12030602_

Round 1

Reviewer 1 Report (Previous Reviewer 1)

As in my first review of this paper, I have the following comments: 

- The meaning of "N-Miss" should be specified at the bottom of the tables

- The English of this paper needs a revision. I suggest the Authors to ask for an English native language speaker revision.

Author Response

As in my first review of this paper, I have the following comments:

- The meaning of "N-Miss" should be specified at the bottom of the tables

Added.

- The English of this paper needs a revision. I suggest the Authors to ask for an English native language speaker revision.

Following your recommendations, we have proceeded to send the manuscript to a professional translator for review and better use of the language.

Reviewer 2 Report (New Reviewer)

Thanks for this study, I appreciate your efforts to quantify abx consumption in a challenging population as that may identify targets for ams activities. Yet, I regret to say that your methods are not clearly described: what was used as a reference tests for ddd, how was the potency defined. Therefore, your results cannot be extrapolated to other settings, which highly limits the usefulness as a measure of neonatal abx consumption.

Author Response

Thanks for this study, I appreciate your efforts to quantify abx consumption in a challenging population as that may identify targets for ams activities. Yet, I regret to say that your methods are not clearly described: what was used as a reference tests for ddd, how was the potency defined. Therefore, your results cannot be extrapolated to other setting, which highly limits the usefulness as a measure of neonatal abx consumption.

Thank you very much for your comment. We agree that the material and methods section was perhaps a bit confusing. That is why we have proceeded to review and modify it in depth, explaining in more detail the entire process carried out and preparing a table that we believe helps to understand the criteria established to define the DDD.

For this reason, we believe that the final DDD obtained after validation in a clinical setting will allow it to be extrapolated to other hospitals around the world.

Reviewer 3 Report (New Reviewer)

Villanueva-Bueno C, et al. propose an interesting topic.

I have some observations that I will divide into sections.

Introduction:  

The clarity of the information presented should be improved. 

Report statistical data on the appropriate and inappropriate use of antimicrobials in neonatal units.

Add information on inappropriate antimicrobial prescribing and its relevance in neonatal units.

It is important to know the effectiveness and validity of different measures in neonatology.

I have reviewed the recent publication of your article (https://doi.org/10.1016/j.eimce.2021.05.012) of Phase I. There are some sections with similar information. I recommend expanding the information depending on the objective of the study.

Materials and Methods

The clarity of the methods used in the Phase II project should be improved.

I have reviewed your published article corresponding to the Phase I study. I recommend adding the type of study, sample size, and how it was calculated.

Results

I recommend improving the clarity of the results.

Correct on line 153: preterm (35.6%), preterm (29.35), and term (35.1%).

Add the percentage of prescriptions belonging to the Intensive Care Unit.

Add the percentage of male neonates.

On line 169. The information presented corresponds to a methodological description. Describe table 3.

Discussion

Present the highlights of your study in terms of your stated objective.

The presentation and comparison of the results of your study with other similar studies should be improved. 

Conclusion

It should be improved based on the objective and the results you found.

Author Response

Villanueva-Bueno C, et al. propose an interesting topic. I have some observations that I will divide into sections.

Introduction:  

The clarity of the information presented should be improved. Report statistical data on the appropriate and inappropriate use of antimicrobials in neonatal units. Add information on inappropriate antimicrobial prescribing and its relevance in neonatal units. It is important to know the effectiveness and validity of different measures in neonatology.

I have reviewed the recent publication of your article (https://doi.org/10.1016/j.eimce.2021.05.012) of Phase I. There are some sections with similar information. I recommend expanding the information depending on the objective of the study.

Thank you very much for your appreciation. We have proceeded to modify the introduction, adding additional information as suggested and providing more relevant data.

We hope that it is now clearer and more complete. However, if you still consider that an additional point is missing, do not hesitate to ask us.

Materials and Methods

The clarity of the methods used in the Phase II project should be improved.

I have reviewed your published article corresponding to the Phase I study. I recommend adding the type of study, sample size, and how it was calculated.

We have proceeded to review and modify the materials and methods section in depth, explaining in more detail the entire process carried out and preparing a table that we believe helps to understand the criteria established to define the DDD.

Results

I recommend improving the clarity of the results.

We have proceeded to reformulate the results and the table where they are exposed to facilitate their understanding.

Correct on line 153: preterm (35.6%), preterm (29.35), and term (35.1%).

Corrected.

Add the percentage of prescriptions belonging to the Intensive Care Unit.

Added.

Add the percentage of male neonates.

Added.

On line 169. The information presented corresponds to a methodological description. Describe table 3.

We have modified the results section, avoiding duplication between the text and the tables.

Discussion

Present the highlights of your study in terms of your stated objective.

The presentation and comparison of the results of your study with other similar studies should be improved. 

We have modified the discussion section. We have added additional information and better correlated the data from our study with other published studies.

Conclusion

 It should be improved based on the objective and the results you found.

We have proceeded to eliminate the sections to better unify the conclusions with the rest of the discussion. We have modified the conclusions so that they are more in line with the objectives of our work and better reflect the importance of our results.

Reviewer 4 Report (New Reviewer)

This is an interesting and important study on the validation of neonatal defined daily dose (DDD) using prescription data from 9 hospitals in Spain. Results and conclusions are potentially valuable for clinical and research purposes. Limitations of the study are adequately described. 

My primary suggestion is that the use of English should be improved throughout the manuscript, especially in the Discussion part. 

Abstract: Page 3, line 19. Complete or reformulate the sentence “Set of 904 prescriptions and 860 …… were analysed.” 

Page 3, line 20. The antimicrobials were mostly prescribed 

Page 8, line 153. Correct: “preterm (35.6%), preterm (29.35)” 

Page 9, line 194. Correct: Discussion instead of Discusion. Page 10, line 227. antimicrobials 

Page 11, lines 242-243: Reformulate the sentence for a correct use of the English language. 

Author Response

This is an interesting and important study on the validation of neonatal defined daily dose (DDD) using prescription data from 9 hospitals in Spain. Results and conclusions are potentially valuable for clinical and research purposes. Limitations of the study are adequately described.

Thank you very much for your thorough review of our manuscript and your positive comments towards our work.

My primary suggestion is that the use of English should be improved throughout the manuscript, especially in the Discussion part.

Following your recommendations, we have proceeded to send the manuscript to a professional translator for review and better use of the language.

Abstract: Page 3, line 19. Complete or reformulate the sentence “Set of 904 prescriptions and 860 …… were analysed.”

Modified

Page 3, line 20. The antimicrobials were mostly prescribed

Corrected

Page 8, line 153. Correct: “preterm (35.6%), preterm (29.35)”

Modified

Page 9, line 194. Correct: Discussion instead of Discusion. Page 10, line 227. antimicrobials

Corrected

Page 11, lines 242-243: Reformulate the sentence for a correct use of the English language.

We have proceeded to modify the phrase so that it is better understood.

Round 2

Reviewer 2 Report (New Reviewer)

Suggest acceptance in the present form.

Reviewer 3 Report (New Reviewer)

The authors corrected the article according to my comments.

This manuscript is a resubmission of an earlier submission. The following is a list of the peer review reports and author responses from that submission.

Round 1

Reviewer 1 Report

Thank you for asking me to review this paper which deals with a topic problem such as the definition of an optimal dose for antibiotics in neonates.

There are very few papers focussing on this topic and the submission is, therefore, relevant.

I would suggest the Authors the following changes:

- The meaning of "N-Miss" should be specified at the bottom of the tables

- I suggest a revision of the English through the manuscript. Verbs are often lacking and the sentences result non always clear

- Even if the design and calculation of neonatal DDD has been performed in a previous referenced study (19), I would suggest the Authors to briefly summarise the information necessary to make the methodology of this study clear to the readers.

Reviewer 2 Report

In this work, the authors aim to validate the Defined Daily Doses for the neonatal population as a new standardised method of antimicrobial consumption in the neonatal population based on a descriptive observational study. The authors claimed that the proposed neonatal Defined Daily Doses seem to be a suitable method to measure antimicrobial consumption in the neonatal population. Nevertheless, the work is routine and does not provide new insight. Specifically, a high similarity in content and methodology was found between this paper and the authors’ recent publication:

Villanueva-Bueno, Cristina, et al. "Antimicrobial defined daily dose in neonatal population." Enfermedades infecciosas y microbiologia clinica (English ed.) 40.2 (2022): 59-65.

In my view, the information revealed herein is not beyond the published work and can’t meet the journal criteria for publication. Therefore, I am not able to recommend the publication of this manuscript in Antibiotics.